# The Comparative Advantage of Cities and Innovation Value Chain: Evidence from China

**Wei Zhao [1], Chao Zhu [1,*] and Yaoyao Zhu [2]**

[1] School of Economics, Zhejiang University, Hangzhou 310058, China; zhaowei2006@zju.edu.cn
[2] School of Education, Zhejiang University, Hangzhou 310058, China; zhuyaoyao1994@zju.edu.cn
* Correspondence: zhuchao1993@zju.edu.cn

**Abstract:** Cities are different in industrial structure; some are specialized while others are diversified. Based on the theory of the innovation value chain, this paper used the innovation data of industrial enterprises from 2008 to 2014 in China to test the comparative advantage of cities in innovation. Our empirical results proved that diversified cities are more suitable for the R&D process of innovation through the labor matching effect and the knowledge spillover effect, while specialized cities are more suitable for the commercialization process through cost-saving effect. Enterprises could choose suitable locations due to their positions in the innovation value chain to achieve sustainable development.

**Keywords:** comparative advantage of cities; diversity; specialization; innovation value chain

## 1. Introduction

China's economy has shifted from a stage of rapid growth to a stage of high-quality development in recent years. The government has implemented several measures to promote economic restructuring and continued sustainable development, among which innovation-driven development is one of the core strategies [1]. For Schumpeter, innovation means introducing a new combination of producing elements and conditions into the production system, which can be presented as new products, new technologies, new markets, new sources of supply, or new forms of organization [2]. This concept suggests that innovation covers the process from the generation of new ideas to commercial applications [3]. From this perspective, the innovation value chain (IVC) theory divides innovation into the following sequential three-phase processes: knowledge production, innovation production, and output production [4,5]. Some researchers also simplified it into the following two sub-processes: the R&D process and the commercialization process [6,7].

Numerous studies focus on successful clusters to demonstrate that proximity enables innovative activities. Most of them investigate the role of geographic, institutional, organizational, cognitive, social, and technological proximities or how they interplay [8–10]. Obviously, innovation has a decidedly geographic dimension, and the role of physical proximity and colocation is pivotal in understanding the dynamics of the innovation process [11]. It is commonly believed that labor and capital are both heavily concentrated in cities, thus cities can offer considerable advantages for innovation [12]. Plenty of evidence shows that innovative activities are even more spatially concentrated in cities than other economic activities [13]. According to the National Innovation Survey Enterprise Database from China, the output of innovative activities is highly agglomerated in eastern developed cities, which accounts for nearly 80 percent (we use the amount of invention patent to measure it). From the perspective of per capita GDP and innovation distribution, it is also clear that cities with a higher degree of industrial agglomeration have absolute advantages in innovation [14,15]. However, if we focus on the most developed cities, the following are some interesting facts (Figure 1): Different cities show different abilities in two processes of innovation. For example, Shenzhen (SZ) is doing the best in the R&D process, but

it is lacking in the commercialization process, while the situation in Zhengzhou (ZZ) is just the opposite. What caused this phenomenon? We know that cities are different in their industrial structure. Some are specialized, while others are diversified. What are the advantages and disadvantages of urban specialization and diversity? Will these result in a division of labor in innovation?

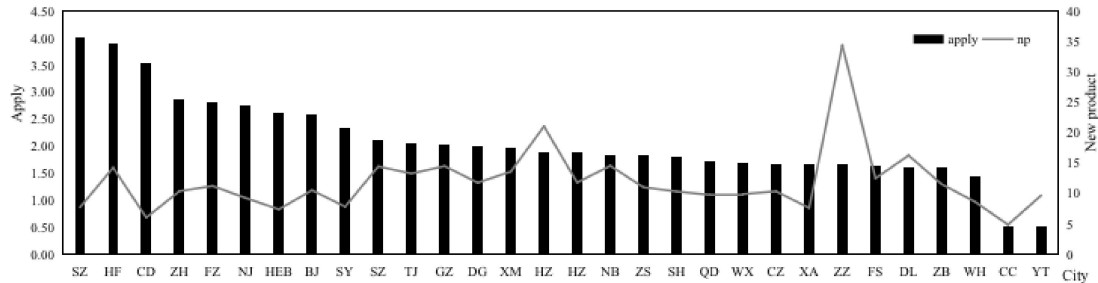

**Figure 1.** Innovation output of cities in China (2014). We selected 30 cities with the highest proportion of manufacturing industry for comparison. The left and right longitudinal axis respectively represent the innovation output of R&D process and commercialization process per 10,000 Yuan R&D investment. Source data are from the National Innovation Survey Enterprise Database (2014).

The topic of economic geography has expanded into mainstream economics as a result of Krugman's several key studies on trade and geography [16,17]. He developed a succinct model to show how a country can endogenously become differentiated into an industrialized "core" and an agricultural "periphery" due to transportation costs, economies of scale, and the share of manufacturing in national income. In fact, geographical proximity also creates a better environment for innovation. Enterprises can benefit from localized competition and share professional labor and business services, which ensures the rapid flow of knowledge [18]. Economists call these benefits from geographical proximity "agglomeration economies", and a lot of empirical evidence can prove it [19–22]. However, it is still controversial whether agglomeration economies are related to the concentration of an industry or to the colocation of different industries in a city [12]. In other words, the debate on the advantages of specialization and diversity.

The comparative advantages of different types of cities are often called "localization economies" and "urbanization economies" [23]. Localization economies are traced back to Marshall [24], its core view is that the massive agglomeration of enterprises in the same industry reduces the flow cost of professional knowledge, and this is conducive to knowledge spillover within the industry. Urbanization economies are linked to the work of Jacobs [25], which holds the view that the diversified environment of a city provides more opportunities for imitation, sharing, and knowledge reorganization. The complementary knowledge exchange across industries will generate stronger spillover. Glaeser et al. summarized the previous research and induced the following three types of agglomeration externalities: MAR's (Marshall-Arrow-Romer) externality emphasizes the innovation and growth effect of local monopolies based on the agglomeration of the same industry; Porter's externality not only emphasizes the positive effect of specialization on knowledge spillover but also the innovation incentive mechanism of local competition; Jacobs' externality summarizes the knowledge spillover effect between different industries [26]. Duranton and Puga think that the combination of localization economies and congestion costs of the city creates "static advantages to urban specialization", while the learning process drawn from local production creates "dynamic advantages to urban diversity" [27].

Whether diversity or specialization better promotes economic growth has been the subject of a heated debate [28,29]. The literature on this topic was surveyed by Groot et al., and they concluded that diversity appears to have a more significant positive effect on economic and productivity growth [30]. However, there are only a few literatures focus on different types of agglomeration economies and innovation directly. Feldman and Audretsch were the first to conduct research on this topic. They found considerable support

for diversity but little support for specialization [31]. After their pioneering research, some researchers also found positive effects of diversity on innovation through different samples and empirical methods, such as Van Oort, Andersson et al., Zhang, and Shi [32–35]. However, some studies show different results, such as Ketelhohn's finding simultaneous positive effects of diversity and specialization. He indicated that diversity is more important in determining the intensity of innovation, while specialization has a stronger role in determining the probability of innovation [36]. Considering the technology intensity of industry, De Beule and Van Beveren believed that specialization is more conducive to innovation for low-tech manufacturing, while diversity is more favorable for high-tech manufacturing [37]. Zhang et al. focused on the regional eco-innovation efficiency (EIE). They found the following nonlinear impacts of specialization and diversity: A U-shaped relationship between the specialization and EIE, and a S-shaped relationship between diversity and EIE [38]. Research in this line of enquiry has also pointed out the role of specialization and diversity in innovation through foreign direct investment (FDI) knowledge spillovers. Ning et al. believe that specialization helps cities absorb FDI knowledge spillovers and diversity also provides a vibrant environment for local innovation [39]. However, Wang et al. thought that specialization diminishes the positive effects of FDI, while a more diversified industrial structure enhances spillovers from FDI [40].

Obviously, there is no consensus on the impact of diversity and specialization on innovation. However, the literature mentioned above does not distinguish the different processes of innovation. According to the theory of innovation value chain (IVC), the R&D process of innovation depends more on the human capital and the externality of knowledge, while the main purpose is to obtain the commercial value of new products in the commercialization process [6,7]. Diversified cities are usually more skill-abundant and better suited for skill-intensive activities [41], whereas more specialized places are better for conducting mass-production of fully developed products with lower costs [27]. Thus, will the relative advantages of diversity and specialization may result in a division of labor in different innovation processes?

To this end, we examine the comparative advantages of diversity and specialization in different processes of innovation based on the IVC. Furthermore, we try to explore the mechanisms of it. Compared with the existing research, this paper may have some contribution to this topic. First, combined with the theory of IVC, we attempt to analyze the relative advantages of localization economies and urbanization economies on innovation from a new line of inquiry. Second, we perform some further mechanism tests that can enhance our understanding of how different agglomeration externalities influence different processes of innovation. Third, we use rich micro-level data on innovative activities of firms, which enables us to discuss the impacts of different agglomeration patterns on micro-level. These may enrich the theoretical basis and empirical evidence of urban agglomeration and innovation, and provide theoretical support for the government to intervene in local industrial agglomeration and guide the location choice of innovative enterprises.

The remainder of this paper is organized as follows: in Section 2, we describe the data source and empirical strategy. Section 3 reports the main empirical findings, and Section 4 tests the mechanisms. Finally, in Section 5, we conclude the key results and give some brief policy implications and discuss the limitations of the paper and directions for future research.

## 2. Data and Empirical Strategy

### 2.1. Data Source and Description

Our empirical study builds on data from the following two main sources: (a) the National Innovation Survey Enterprise Database (NISED), and (b) the China City Statistical Yearbook (CCSY). The NISED provides detailed information on various scientific and data of innovative activities of industrial enterprises from 2008 to 2014. It is one of the most comprehensive and important databases for studying the innovative activities of micro-

enterprises in China. In addition, the CCSY records the main economic data of China's cities over the years, which will help us calculate relevant indicators.

To capture a firm's ability in different processes of innovation, we use the number of applications of invention patents to measure the output of R&D process and the sales revenue of new products to measure the output of the commercialization process. For input, we use the number of R&D people and the amount of internal and external R&D expenditures.

The core independent variable of this paper is the comparative advantage of cities, or the different types of agglomeration economies. Cities are different in industrial structures; some are specialized while others are diversified. We use one minus the Herfindahl–Hirschman index (HHI) to measure the type of cities, which is given by the following:

$$TC_i = 1 - \sum_j s_{ij}^2 \tag{1}$$

where $s_{ij}$ is the share of employment of industry $j$ in city $i$. If the economic activities in the city are fully concentrated in one sector, which means the city is fully specialized, we will find $TC_i = 0$, and this index increases as activities in this city become more diverse.

In order to avoid the problem of missing variables, we choose both the characteristic variables of enterprises and the location cities under the condition of ensuring the exogenous. The descriptions of all the variables are presented in Table 1.

**Table 1.** Variable descriptions.

| Theme | Variable | Symbol | Definition |
|---|---|---|---|
| Dependent variable | R&D output of innovation | RDOI | Number of applications of invention patent |
| | Commercialization output of innovation | COI | Sales revenue of new products (1000 yuan) |
| Core independent variable | Type of cities | TC | One minus the HHI |
| | Human capital input | HCI | Number of people in R&D |
| | R&D input | RDI | Amount of internal and external R&D expenditures |
| Control variable (enterprise characteristics) | Technology import expenditure | TIE | The expenditure of technology import both foreign and domestic (1000 yuan) |
| | Degree of internationalization | DI | Proportion of exports in sales revenue of new products |
| | Industry leadership | IL | Number of national or industrial standards |
| | High-degree employee | HE | Proportion of highly educated employees |
| | female staff | FS | Proportion of female staff |
| | Government support | GS | Tax reduction and exemption |
| | urban scale | US | Total population |
| | Economic development level | EDL | GDP per capita (10,000 yuan) |
| Control variable (urban characteristics) | Financial market scale | FMS | Balance of deposits and loans of financial institutions (10,000 yuan) |
| | Internet construction level | ICL | Internet access rate |
| | Local expenditure on education | LEE | Proportion of local financial expenditure on education |

### 2.2. Dealing with Endogeneity

One reason why local determinants of agglomeration economies can be endogenous is that some missing variables determine them simultaneously with the local outcome [42]. In order to cope with this issue, we control both firm-fixed effects and time-fixed effects.

Another alternative strategy for coping with endogeneity is to find instruments that can deal with both reverse causality and missing amenities. Historical instruments are used to measure the long-lagged values of agglomeration variables. Historical values will have an inertial impact on the local population and employment structure, but current location selection or innovative activities cannot be related to historical characteristics. Therefore, as long as the lags are long enough, instruments are believed to be exogenous [43,44]. Considering the different classification standards of industry in the urban statistical yearbook before and after 2003 (there are 15 two-digital industries before 2003 while 19 industries after 2003), in order to ensure the accuracy of the results, we selected historical variables in 2003 as the first instrumental variable.

The third strategy we use is to construct a Bartik instrumental variable. The core idea is to simulate the estimated value of each period of the sample by using the initial share and overall growth rate of the analysis unit. The estimated value is highly correlated with the actual value, but not with other residual terms, which conforms to the relevance and exogeneity. Referring to the research of Ottavian and Peri [45], Kemeny and Storper [46],

we use "shift-share design" to build the Bartik instrumental variable. Specifically, we take 2008 as the base period ($t_0$), and calculate the growth rate of employees in each industry in simple cities from 2009 to 2014, then estimate the employees over the years ($e_{ijt}$), this leads to the following:

$$e_{ijt} = \left(E_{jt}/E_{jt_0}\right) \cdot e_{jt_0} \tag{2}$$

Finally, we use the estimated value and formula (1) to calculate the degree of diversification of cities from 2009 to 2014 and obtain the Bartik instrumental variable of agglomeration economies. Obviously, this instrumental variable is highly correlated with the actual degree of city diversification, but not with other residual terms that will affect urban innovation. Therefore, the endogenous problem can be well solved.

### 2.3. The Empirical Model

From the literature on innovation and technological change [47], the model of the knowledge production function of an enterprise can be represented as follows:

$$I_k = \alpha RD_k^\beta HK_k^\gamma \varepsilon_k \tag{3}$$

where *I* stands for the degree of innovative activity, *RD* represents R&D inputs, and *HK* represents human capital inputs.

Combined with the above model and the research of Feldman and Audretsch [31], our basic empirical model is constructed as follows:

$$I_{kt} = \beta_0 + \beta_1 TC_i + \beta_X X + \mu_k + \nu_t + \varepsilon_{kt} \tag{4}$$

where $I_{kt}$ represents the innovation output of enterprise *k* in year *t*, including the output of R&D and commercialization process. $TC_i$ represents the type of city where the enterprise is located. *X* stands for other control variables, including characteristics of enterprises and location cities. Additionally, $\mu_k$, $\nu_t$ stands for individual and time-fixed effects, respectively, $\varepsilon_{kt}$ stands for random error term.

## 3. Empirical Estimates

### 3.1. Baseline Regression

This section provides an answer to the following first two research questions: What are the advantages and disadvantages of urban specialization and diversity? Will this result in a division of labor in innovation? Table 2 presents the baseline estimates of our empirical model. Columns (1) and (3) include only the core independent variable (Type of Cities) and innovation input variables; columns (2) and (4) add controls. All models control the two-way fixed effects of individuals and time. Regression results reveal a positive relationship between the type of cities and the R&D output of innovation and a negative relationship between the type of cities and the commercialization output of innovation. These effects are highly statistically significant, which means diversified cities are more suitable for the R&D process of innovation, while specialized cities are more suitable for the commercialization process.

### 3.2. Instrumental Variables

Table 3 shows the regression results of the two-stage least square method with the instrumental variables we introduced in 2.2. The results of the first step indicate that the instrumental variables are highly correlated with the endogenous variables; both have a high correlation coefficient of up to 0.75, which meets the requirement of relevance. At the same time, we perform some further tests to determine whether the instruments can pass the weak identification test and the weak identification test. The results show the rationality of our choice of instruments. In addition, the results of the second step indicate that no significant change has taken place in the coefficient of the core independent variable. Thus, our conclusion is credible.

**Table 2.** The comparative advantage of cities and innovation value chain.

| Variables | (1) | (2) | (3) | (4) |
|---|---|---|---|---|
| | **R&D Output of Innovation** | | **Commercialization Output of Innovation** | |
| Type of cities | 0.1678 *** (0.0382) | 0.1337 * (0.0759) | −0.5673 *** (0.2507) | −0.2729 *** (0.1342) |
| Human capital input | 0.1018 *** (0.0047) | 0.0867 *** (0.0091) | 0.5482 *** (0.0294) | 0.1005 *** (0.0183) |
| R&D input | 0.0435 *** (0.0028) | 0.0419 *** (0.0063) | 0.5776 *** (0.0181) | 0.3364 *** (0.0141) |
| Technology import expenditure | | 0.0042 *** (0.0013) | | 0.0133 *** (0.0023) |
| Degree of internationalization | | 0.0204 (0.0211) | | −0.0963 ** (0.0432) |
| Industry leadership | | 0.0909 *** (0.0093) | | 0.0406 *** (0.0150) |
| High-degree employee | | 0.1429 *** (0.0347) | | −0.0276 (0.0648) |
| Female staff | | 0.0642 * (0.0344) | | 0.1930 *** (0.0673) |
| Government support | | 0.0102 *** (0.0015) | | 0.0145 *** (0.0025) |
| Urban scale | | −0.0367 (0.0263) | | 0.0509 (0.0523) |
| Economic development level | | 0.0031 (0.0136) | | −0.0070 (0.0215) |
| Financial market scale | | 0.0006 (0.0065) | | −0.0340 *** (0.0123) |
| Internet construction level | | −0.0107 (0.0229) | | 0.0791 ** (0.0403) |
| Local expenditure on education | | 0.2236 (0.1556) | | −0.4663 (0.2915) |
| Constant | −0.2339 *** (0.0358) | 0.0308 (0.2356) | 0.6051 *** (0.2349) | 7.2598 *** (0.4108) |
| Individual fixed effects | Yes | Yes | Yes | Yes |
| Time fixed effects | Yes | Yes | Yes | Yes |
| Number of observations | 235,711 | 92,058 | 238,881 | 92,103 |
| Adjusted-$R^2$ | 0.6268 | 0.6657 | 0.6290 | 0.7830 |

Notes: Standard errors, clustered by enterprise, in parentheses. ***, **, and * refer to the 1, 5, and 10 percent significance level respectively.

**Table 3.** Regression results with instrumental variables.

| | **First Step** | | | |
|---|---|---|---|---|
| Variables | (1) | (2) | (3) | (4) |
| | **Types of Cities** | | | |
| Historical instrument variable | 0.7671 *** (0.0248) | | 0.7671 *** (0.0248) | |
| Bartik instrument variable | | 0.7508 *** (0.0088) | | 0.7508 *** (0.0088) |
| | **Second Step** | | | |
| Variables | (1) | (2) | (3) | (4) |
| | **R&D Output of Innovation** | | **Commercialization Output of Innovation** | |
| Type of Cities | 0.1558 *** (0.0378) | 0.1247 *** (0.0072) | −0.2044 *** (0.0050) | −0.2224 *** (0.0112) |
| Kleibergen-Paap rk LM statistic | 302.74 *** | 437.49 *** | 302.74 *** | 437.49 *** |
| Cragg-Donald Wald F statistic | 6499.02 *** | 12,482.2 *** | 6499.02 *** | 12,482.2 *** |
| Control variables | Yes | Yes | Yes | Yes |
| Number of observations | 92,058 | 92,058 | 92,103 | 92,103 |

Notes: Standard errors, clustered by enterprise, in parentheses. *** refers to the 1 percent significance level.

### 3.3. Robustness Checks

We perform the following three sets of additional analyses to evaluate the robustness and credibility of our results: (i) replace the methods of measuring the output of innovation,

(ii) consider the lag effect of innovation investment, (iii) use Tobit model to deal with zero values flaw, (iv) consider the spatial autocorrelation effect.

Considering scientific papers also reflect the R&D process, we use the sum of invention patents and papers to replace the output of R&D. Besides, we also use the output value of new products to replace the output of commercialization. Columns (1) and (2) in Table 4 show that the value, sign, and significance of coefficients of the independent variable almost have no change after replacing the methods of measuring the dependent variables.

**Table 4.** Robustness checks: methods (i)–(iii).

| Variables | (1) | (2) | (3) | (4) | (5) | (6) |
|---|---|---|---|---|---|---|
| | Replace Dependent Variables | | Lag Effect | | Tobit Model | |
| Type of cities | 0.2194 ** | −0.2681 *** | 0.2198 *** | −0.1187 *** | 0.3851 *** | −0.3443 *** |
| | (0.0969) | (0.0175) | (0.0825) | (0.0038) | (0.0494) | (0.0557) |
| Control variables | Yes | Yes | Yes | Yes | Yes | Yes |
| Individual Fixed effects | Yes | Yes | Yes | Yes | Yes | Yes |
| Time fixed effects | Yes | Yes | Yes | Yes | Yes | Yes |
| Number of observations | 92,029 | 92,072 | 60,112 | 60,585 | 107,566 | 107,614 |
| Adjusted-$R^2$ | 0.7146 | 0.6847 | 0.6527 | 0.3997 | | |

Notes: Standard errors, clustered by enterprise, in parentheses. *** and ** refer to the 1 and 5 percent significance level respectively.

Sometimes, the impact of current innovation investment or other external factors may not be reflected until later, so we also consider the lag effect of innovation. We use the lag of enterprise innovation output of the two processes for one period as the dependent variable to retest our model. Columns (3) and (4) report the results of lag models. Additionally, empirical findings remain statistically significant.

To take into account the zero value flaw in our sample, we perform an additional robustness test by using the Tobit model. The results in columns (5) and (6) show that there is no significant change in the main coefficients. Thus, our conclusion is very robust.

Previous studies have pointed out that innovative activities are spatially dependent [48,49]. Therefore, it is necessary to introduce a spatial regression model that considers spatial-related factors. It is quite difficult to test the spatial correlation under firm-level, we aggregate the data at the city level and only use control variables of city characteristics. Then we use the spatial Durbin model (SDM), which could consider the spatial correlation of dependent and independent variables simultaneously and reconstructs our empirical model as follows:

$$I_{it} = \beta_0 + \beta_1 TC_i + \beta_{X'} X' + \rho W I_{it-1} + \gamma_1 WTC_i + \gamma_{X'} W X' + \mu_i + \nu_t + \varepsilon_{it} \tag{5}$$

where $W$ stands for the spatial weight matrix, $\rho$, $\gamma_1$, $\gamma_{X'}$ represents the spatial autoregressive coefficient of dependent and independent variables, respectively.

We use the actual geographical distance and economic distance constructs in the spatial weight matrix, respectively, as follows [50]:

$$W_d = \begin{cases} \frac{1}{d^2}, & m \neq n \\ 0, & m = n \end{cases} \tag{6}$$

$$W_e = \begin{cases} \frac{1}{|\overline{Y_m} - \overline{Y_n}|}, & m \neq n \\ 0, & m = n \end{cases} \tag{7}$$

In expression (6), $d$ represents the geographical distance between cities $m$ and $n$, in expression (7), $\overline{Y}$ represents the GDP of each city.

Applying the SDM under the weight of geographical and economic distance, our empirical results are presented in Table 5. We can see that the coefficient of spatial lag term ($\rho$) of each model is significantly positive, which means the innovation of cities will be positively affected by neighborhoods and cities with similar economic development levels. In addition, there is no significant change of the coefficient in different types of cities.

Moreover, we found the diversity of geographically adjacent cities may cause a negative effect on the output of the R&D process, which shows a degree of backwash effect.

**Table 5.** Robustness checks: (iv) SDM.

| Variables | (1) | (2) | (3) | (4) |
| --- | --- | --- | --- | --- |
| | Geographical Weight Matrix | | Economic Weight Matrix | |
| ρ | 0.4511 *** | 0.4231 ** | 0.3553 *** | 0.3312 *** |
| | (0.0784) | (0.2552) | (0.0699) | (0.0671) |
| Type of cities | 0.3378 *** | −0.4581 *** | 0.2966 *** | −0.2167 *** |
| | (0.0589) | (0.0075) | (0.0745) | (0.0125) |
| W × Type of cities | −0.0622 *** | 0.0284 | 0.0137 | 0.0135 |
| | (0.0083) | (0.0368) | (0.0143) | (0.0145) |
| Control variables | Yes | Yes | Yes | Yes |
| Individual fixed effects | Yes | Yes | Yes | Yes |
| Time fixed effects | Yes | Yes | Yes | Yes |
| Number of observations | 1985 | 1985 | 1985 | 1985 |
| Adjusted-$R^2$ | 0.8236 | 0.8157 | 0.7996 | 0.7838 |

Notes: Standard errors, clustered by enterprise, in parentheses. *** and ** refer to the 1 and 5 percent significance level respectively.

### 3.4. Heterogeneity across Different Kinds of Industries and Ownership

In order to gather additional insights about the impact of different types of cities on firms' innovation, we perform heterogeneity analysis from two aspects.

First, we distinguish our sample enterprises from different kinds of industries. According to the Statistical Classification Catalogue of High-tech Industries from the National Bureau of Statistics of China, we divide the sample into high-tech enterprises and non-high-tech enterprises for analysis, respectively. Table 6 reports the results. It can be seen that the type of city only affects the high-tech enterprises' innovation activities. This conclusion is consistent with Yang et al. and Zhang et al. that agglomeration externality is more important for the innovation of the high-tech industry [21,51].

**Table 6.** Heterogeneity across different kinds of industries.

| Variables | (1) | (2) | (3) | (4) |
| --- | --- | --- | --- | --- |
| | High-Tech Enterprise | | Non-High-Tech Enterprise | |
| Type of cities | 0.1882 ** | −0.5673 *** | −0.4535 | 0.5110 |
| | (0.0793) | (0.2507) | (0.2954) | (0.6162) |
| Control variables | Yes | Yes | Yes | Yes |
| Individual fixed effects | Yes | Yes | Yes | Yes |
| Time fixed effects | Yes | Yes | Yes | Yes |
| Number of observations | 81,991 | 82,031 | 9433 | 9439 |
| Adjusted-$R^2$ | 0.6678 | 0.7877 | 0.7403 | 0.8059 |

Notes: Standard errors, clustered by enterprise, in parentheses. *** and ** refer to the 1 and 5 percent significance level respectively.

We also consider the heterogeneity across different ownerships. Some studies show that due to different property rights and incentive mechanisms, state-owned enterprises lack competitiveness in innovation activities compared with private enterprises and foreign-funded enterprises [52]. We regard those whose state capital accounts for more than half of the paid in capital as state-owned enterprises, while others are non-state-owned enterprises. The regression results in Table 7 show that our previous conclusions are only applicable to non-state-owned enterprises. The possible reason is that state-owned enterprises tend to engage in more basic research activities, while the investment and risk of such activities are higher, the return is not as high. Therefore, market factors such as agglomeration

economies may not determine the innovation decisions of state-owned enterprises. At this time, state-owned enterprises have become an important tool for the government to solve market failure.

**Table 7.** Heterogeneity across different ownership.

| Variables | (1) | (2) | (3) | (4) |
|---|---|---|---|---|
| | State-Owned Enterprise | | Non-State-Owned Enterprise | |
| Type of cities | 0.0903 | 0.0189 | 0.2821 *** | −0.2552 * |
| | (0.0795) | (0.5619) | (0.0233) | (0.1407) |
| Control variables | Yes | Yes | Yes | Yes |
| Individual fixed effects | Yes | Yes | Yes | Yes |
| Time fixed effects | Yes | Yes | Yes | Yes |
| Number of observations | 12,770 | 12,775 | 78,948 | 78,987 |
| Adjusted-$R^2$ | 0.7881 | 0.8359 | 0.6284 | 0.7671 |

Notes: Standard errors, clustered by enterprise, in parentheses. *** and * refer to the 1 and 10 percent significance level respectively.

## 4. Mechanisms

The previous section has shown that diversified cities are more suitable for the R&D process of innovation, while specialized cities are more suitable for the commercialization process. In this section, we will test the mechanisms of urban comparative advantages in different innovation processes.

### 4.1. Advantages of Diversity on R&D Process

Different from innovation in business organization methods and marketing means, technological innovation depends more on human capital and the externality of knowledge. Therefore, the success of enterprise technological innovation depends on whether it can match the appropriate labor force and make use of knowledge spillover, while diversified cities do have comparative advantages in these aspects.

4.1.1. Labor Matching Effect

In reality, small and medium-sized cities tend to focus on a few industries and are usually more specialized. While big cities usually include many industries that are not directly related and tend to be more diversified [53]. Therefore, the types of labor in diversified cities are more abundant, and enterprises facing technological innovation often choose large cities to improve the probability of matching with corresponding skilled labor [41]. In addition, technological innovation often requires higher labor skills. Compared with small and medium-sized cities, the cost of living in large cities is also higher, which forces some low-skilled labor to move out. This "spatial self-selection" mechanism makes the proportion of highly skilled labor in diversified cities becomes higher, which reduces the search cost and mismatch probability of enterprises and finally promotes the improvement of technological innovation efficiency of enterprises [54].

Therefore, we introduce labor matching quality as the mechanism variable. We use the proportion of university students in a city to measure the quality of labor matching; the data is taken from the China City Statistical Yearbook. We add the interactive items of type of cities and labor matching quality into our baseline model. Column (1) in Table 8 shows that the coefficient of the interactive item is significantly positive. This proves that diversity can promote R&D innovation through labor matching effect.

**Table 8.** Advantages of diversification on R&D process.

| Variables | (1) | (2) |
|---|---|---|
| | **R&D Output of Innovation** | |
| Type of cities | 0.1080 ** (0.0784) | 0.0945 ** (0.0079) |
| Type of cities × Labor matching quality | 0.0009 *** (0.0002) | |
| Type of cities × Knowledge spillover | | 0.0876 *** (0.0094) |
| Control variables | Yes | Yes |
| Individual fixed effects | Yes | Yes |
| Time fixed effects | Yes | Yes |
| Number of observations | 94,275 | 95,452 |
| Adjusted-$R^2$ | 0.6828 | 0.6816 |

Notes: Standard errors, clustered by enterprise, in parentheses. *** and ** refer to the 1 and 5 percent significance level respectively.

### 4.1.2. Knowledge Spillover Effect

The diversity of knowledge and skill is an important element of innovation [55]. The exchange and collision of differentiated knowledge is one of the important ways to produce new knowledge [56]. If the industrial structure of a city is more diversified, the mobility of labor among industries is also higher. When workers change jobs across industries, the combining of their past skills and knowledge with new jobs will produce incremental knowledge. At this time, diversified cities can provide more communication opportunities for heterogeneous labor so as to promote knowledge spillover.

We introduce knowledge spillover as our mechanism variable, which is measured by amount of invention patent applications of the whole city, our data is from the China's State Intellectual Property Office. From column (2) of Table 8, we can see that the interactive item of type of cities and knowledge spillover is significantly positive, which means diversified cities are more suitable for the R&D process of innovation because of the knowledge spillover effect.

### 4.2. Advantages of Specialization on Commercialization Process: Cost Saving Effect

In the commercialization process of innovation, the main purpose of enterprises is to obtain the commercial value of new products. It is important to reduce production costs and promote sales capacity. MAR externality presents that specialization can form a professional "labor pool", and enterprises can hire labor with professional knowledge of the industry more effectively, which can help them reduce the employment cost [26]. At the same time, in specialized cities, final product manufacturers can share a large number of intermediate product suppliers to reduce input costs and obtain competitive advantages [57]. Therefore, enterprises engaged in the mass production of new products are more willing to choose specialized cities.

We use the cost-income ratio of the main business of the enterprises to measure the operating cost as our mechanism variable. After adding the interactive items of type of cities and operating cost into the baseline model, the result in Table 9 shows that the coefficient of the interactive item is significantly positive. Therefore, enterprises in specialized cities usually have lower production costs in the process of commercialization.

**Table 9.** Advantages of specialization on commercialization process.

| Variables | (1) |
|---|---|
| | **Commercialization Output of Innovation** |
| Type of cities | −0.2391 *** |
| | (0.0579) |
| Type of cities × Operating cost | 0.1444 *** |
| | (0.0025) |
| Control variables | Yes |
| Individual fixed effects | Yes |
| Time fixed effects | Yes |
| Number of observations | 72,051 |
| Adjusted-$R^2$ | 0.8026 |

Notes: Standard errors, clustered by enterprise, in parentheses. *** refers to the 1 percent significance level.

## 5. Discussion and Conclusions

Urban agglomeration has become one of the key drivers of economic growth and technology innovation, but whether diversity or specialization better promotes the innovation performance of enterprises has not reached a consensus yet [30]. Our study investigated this issue through a new perspective on the innovation value chain. We divided the innovative activities of enterprises into the following two independent processes: R&D and commercialization. The different agglomeration economies may have different impacts on each process of innovation, and enterprises can choose their location according to their innovation strategies.

Our main results can be summarized as follows: First, we found credible evidence that diversified cities are more suitable for the R&D process of innovation, while specialized cities are more suitable for the commercialization process. This conclusion is quite consistent with reality. A wide diversity of local service options and skilled labor allows enterprises to better match their various needs in the R&D process, which increases the probability of setting up R&D labs in this city [58,59]. While disaggregation of R&D and commercialization processes has induced the formation of secondary markets in disembodied technology inputs, at this time, specialization can minimize the commercialization costs of innovation [60]. Then the phenomenon we observed in the introduction can be well explained. In China, Shenzhen is a special economic zone with a variety of industries, which makes it a global center of technology innovation and attracts development hubs of high-tech companies such as Huawei, Tencent, and Baidu. Additionally, Zhengzhou, famous for being called "the world's mobile phone factory" (such as Foxconn Science and Technology Zone), is a traditional manufacturing industry strong city. The advantage of dense specialized labor here is that it helps enterprises reduce their costs of commercialization significantly.

Second, we tested how localization economies and urbanization economies work. Diversified cities tend to have more skilled labor, which improves the quality of matches between employers and employees. Besides, successful invention efforts in one industry may have positive effects on other industries in the same city [61]. Diversified cities usually possess more invention patents in various industries, which will promote their R&D performance through a knowledge spillover effect. While located in a more specialized city, enterprises can hire labor with the required professional knowledge of the industry more effectively and share a large number of intermediate product suppliers, which will help them reduce the cost of the production of new products.

Different resource endowments, geographical environments, and policies of cities determine their different industrial structures. The industries of diversified cities are "large and complete", while those of specialized cities are "small but professional", which forms their comparative advantages. Enterprises engaging in innovation should choose suitable locations due to their positions in the innovation value chain to achieve sustainable development. While the findings presented here may not be suitable for enterprises and cities outside China, they do provide a new perspective to examine the comparative

advantage of cities in innovation, and support a call for much more detailed and in-depth research into this issue. Perhaps some detailed contrastive comparisons of individual cities would help to reveal the causal stories behind divergent patterns [29].

Still, there are some issues that deserve further investigation. A growing community of economic geographers has invoked the notions of "related variety" and "unrelated variety" to analyze regional variations in growth [62]. The concept was first introduced by Frenken et al., who divided the Jacobs externalities into two kinds according to whether two industries share some cognitive structures, and they expected that knowledge spillover within the region would occur primarily among related sectors, and only to a limited extent among unrelated sectors [63]. However, some researchers believe that unrelated knowledge combinations may be the unexplored potential for regional industrial path development [62]. Thus, it is necessary to figure out the importance of related and unrelated varieties on regional innovation in the future. In addition, a few scholars have made some preliminary attempts, such as Barbieri et al., who analyzed whether related and unrelated varieties matter for the development of green technology, and found that unrelated varieties are the main drivers of green technology development in early stages, while related varieties more prominent as the technology enters into maturity [64].

**Author Contributions:** Conceptualization, W.Z. and C.Z.; methodology, C.Z.; software, C.Z.; validation, C.Z. and Y.Z.; formal analysis, C.Z.; writing—original draft preparation, C.Z.; writing—review and editing, W.Z. and Y.Z.; All authors have read and agreed to the published version of the manuscript.

**Funding:** This research was funded by the key base for research on humanities of Ministry of Education of China, grant number 16JJD790050.

**Institutional Review Board Statement:** Not applicable.

**Informed Consent Statement:** Not applicable.

**Data Availability Statement:** The case analysis data used to support the findings of this study are available from the corresponding author upon request.

**Conflicts of Interest:** The authors declare no conflict of interest.

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
