# Peer review of "The Comparative Advantage of Cities and Innovation Value Chain: Evidence from China"

_sustainability, doi:10.3390/su14063510_

Round 1

Reviewer 1 Report

The article examines the sources of comparative advantage of cites related to innovation output of enterprises. This scope intersects fields of economic geography, urban economics and innovation studies. It touches the basics of urban economic development and provides a takeaway for the broader discussion on spatial diversification of wealth and poverty, thus providing practical solutions for development policies. 

I have the following remarks:

1) Thinking about localisation and urbanisation economies is thinking about geographical proximity. However, there are alternative concepts of proximity supporting innovativeness. It is worth mentioning in the Introduction.

2) The references not fulfilled the literature body relevantly. The direct linkages to Jacobs, Marshall-Arrow-Romer and Porter externalities are not readable. There is no reference to Porter's and Krugman's seminal works. Only five from 25 references were published recently (2015 and later), while the discussion is still vital. Therefore, the set of literature referred to is partially outdated. 

3) The potential spatial autocorrelation bias needs explanation. Are the methods employed in the study are robust in these terms?

4) There is no broader interpretation of results in the context of previous studies,  contribution to the science and further research directions in the text. The discussion needs to be improved fundamentally.

5) The issue requires stronger links to the sustainability concept. 

Reviewer 2 Report

  1. The content could be better contextualized with respect to the theoretical background.
  2. In the Introduction, please review the recent articles on your topic. Additionally, please show your original contribution. 
  3. Please improve the style of the English language.
  4. In the Discussions, please show the limits of your research with respect to other research on your topic. 

Round 2

Reviewer 1 Report

All my comments have been addressed. The changes and additions meet my expectations. I have no further comments.

Wish you luck!

Reviewer 2 Report

The paper is improved. I have no more comments.